# Colonization in Artificial Seaweed Substrates: Two Locations, One Year

Diego Carreira-Flores [1,*] , Regina Neto [1] , Hugo R. S. Ferreira [2,3] , Edna Cabecinha [4] , Guillermo Díaz-Agras [5] , Marcos Rubal [1] and Pedro T. Gomes [1]

1 Centre of Molecular and Environmental Biology (CBMA)/Aquatic Research Network (ARNET), Department of Biology, University of Minho, 4704-553 Braga, Portugal; marcos.rubal@bio.uminho.pt (M.R.)
2 Centro de Estudos do Ambiente e do Mar (CESAM), Departamento de Biologia, Universidade de Aveiro, Campus Universitário de Santiago, 3810-193 Aveiro, Portugal
3 Tour du Valat, Research Institute for the Conservation of Mediterranean Wetlands, 13200 Le Sambuc, France
4 Centre for the Research and Technology of Agro-Environmental and Biological Sciences (CITAB), Inov4Agro, Department of Biology and Environment, University of Trás-os Montes and Alto Douro, 5000-801 Vila Real, Portugal
5 Estación de Bioloxía Mariña da Graña, University of Santiago de Compostela, Rede de Estacións Biolóxicas da USC (REBUSC),15590 Ferrol, Spain
* Correspondence: diego.carreira@bio.uminho.pt

**Abstract:** Artificial substrates have been implemented to overcome the problems associated with quantitative sampling of marine epifaunal assemblages. These substrates provide artificial habitats that mimic natural habitat features, thereby standardizing the sampling effort and enabling direct comparisons among different sites and studies. This paper explores the potential of the "Artificial Seaweed Monitoring System" (ASMS) sampling methodology to evaluate the natural variability of assemblages along a coastline of more than 200 km, by describing the succession of the ASMS' associated macrofauna at two Rías of the Galician Coast (NW Iberian Peninsula) after 3, 6, 9, and 12 months after deployment. The results show that macrofauna assemblages harbored by ASMS differ between locations for every type of data. The results also support the hypothesis that succession in benthic communities is not a linear process, but rather a mixture of different successional stages. The use of the ASMS is proved to be a successful standard monitoring methodology, as it is sensitive to scale-dependent patterns and captures the temporal variability of macrobenthic assemblages. Hence, the ASMS can serve as a replicable approach contributing to the "Good Environmental Status" assessment through non-destructive monitoring programs based on benthic marine macrofauna monitoring, capturing the variability in representative assemblages as long as sampling deployment periods are standard.

**Keywords:** epifaunal assemblages; artificial substrates; artificial seaweed monitoring system; succession

## 1. Introduction

The evaluation of the natural variability of assemblages is essential when assessing and monitoring the natural succession processes in the marine environment, as well as when quantifying the scale of anthropogenic impacts [1–3]. Environmental and biological processes shape assemblages' structure at different spatial and temporal scales [4,5]. Consequently, studies on assemblages' temporal and spatial patterns are required in order to understand these succession patterns [6], constituting a fundamental key step when designing monitoring protocols. The monitoring of marine assemblages is an essential tool in environmental management, as envisaged by the European Marine Strategy Framework Directive (MSFD; 2008/56/EC), with species composition being a basic descriptor for evaluating 'Good Environmental Status' (GES). Still, scientific uncertainties regarding benthic processes and the difficulty of performing sampling and monitoring make assessing what

constitutes GES regarding sea-floor integrity a challenging task [7,8]. This assessment of GES becomes even more complex when the study sites are located within protected marine areas, where the use of non-destructive sampling methods is mandatory.

Marine benthic monitoring is undertaken in three different scenarios: soft bottoms, hard bottoms, and mixtures of the two. Marine soft bottoms' highly diverse benthic macrofaunal communities [9] are relatively easy to sample using standard quantitative sampling devices such as grabs or corers [10–12]. For these well-studied ecosystems, indexes have been developed to assess the ecological quality of the surrounding marine environment [8,13]. Conversely, rocky bottoms present particular difficulties for standard surveys of biodiversity because of their spatial complexity and variability. The natural and physical heterogeneity of rocky reefs, such as the presence of sediments, large macroalgae settlements, rocky crevices, and the presence of biogenic structures such as mussels beds, conditions macrofauna assemblages [14–16]. Rocky reefs also support diverse assemblages of species from many phyla of invertebrates [17,18]. Heterogeneity directly influences the structure of the assemblages by creating new habitats, microenvironments, and resources. Additionally, it indirectly affects the intensity of biotic processes such as predation and competition by altering them [6]. Thus, sampling these habitats can be challenging due to their heterogeneity and complexity, which raises several questions about the most effective methods for capturing the spatial variability of these communities [11,19].

Natural assemblages are frequently complex and irregular along reduced spatial extension and/or time scales, often mainly driven by the availability of space [6] and the presence of disturbances [20,21]. The colonization process starts with the arrival of early colonizing species, with a given set of life-history characteristics. These early colonizing species have particular ecological features that influence the settlement of late-colonizing species, potentially leading to various succession processes based on the interactions between early and late colonizers [1,22]. Intense disturbances create new opportunities for settlement by providing additional uncolonized space, occasionally resulting in the colonization process being restarted [6,23]. The whole process of colonization and succession is continuous and traceable over different time scales (e.g., hours, days, months, or years). Hence, benthic communities can be understood to be a mixture of different successional stages, rather than as a linear process [24,25]. Nevertheless, the mechanisms of colonization and succession are not clearly understood, and the patterns of successional changes are not yet well documented [18], especially in NW Iberia.

Benthic invertebrates are reliable key biological elements for monitoring programs because they are able to reflect the state of environmental quality [26], as they are conspicuous, easily sampled, and respond quickly to changes [27,28]. Artificial substrates (AS) have been implemented to overcome problems with the quantitative sampling of benthic macrofauna in structurally complex environments [29–34] due to their success as collectors of macrofauna [35–37]. Artificial substrates are regarded as a viable non-destructive alternative, provided that the natural structural complexity is mimicked by the AS [21,29,31,35,38–41]. They have also been proved to be a valid tool for distinguishing macroalgae-like and crevice-like macrofauna assemblages from distinctive locations and over different time scales [18,38,41].

Artificial substrates provide uncolonized habitats that mimic the essential features of the natural habitat, standardizing the sampling effort and enabling direct comparisons between different sites and studies [19,21,29,42]. With known features and the ability to be developed with a deliberated target, AS are adequate for standardizing sampling devices for quantitative non-destructive sampling [19,21,32,34]. While standardizing deployment time and duration is crucial for obtaining comparable data in monitoring programs, there is variability in the literature regarding the periods of deployment for artificial substrates (AS). Some authors have utilized deployment periods ranging from a few days [34,39] to one month [29], of several months [18,39], or even lasting multiple years [19,42,43]. In certain cases, a standardized 3-month deployment period has been implemented to collect

macrofaunal data using AS [44], but only a limited number of studies have examined the colonization process within this specific 3-month period [21].

A successful standard monitoring methodology based on AS must be sensitive to scale-dependent patterns and capture the variability of macrobenthic communities over time. This entails sampling the seasonal changes and fluctuations derived from the natural ecological variability of the deployment sites, especially if they have similar features. Thus, standard methodologies must guarantee their ability to represent the natural variability over small and large spatial scales, without simply being colonized by some opportunistic species that does not vary over time, leading the methods to not be representative of the surrounding natural assemblages. The process of validation of the "Artificial Seaweed Monitoring System" (ASMS) standard sampling methodology proposed by Carreira-Flores et al. [38,41] went through the standardization of the deployment (date and period) and the validation of the possibilities for capturing natural variability over time at different spatial scales.

Understanding succession patterns is a fundamental key step when designing monitoring protocols, and the knowledge of the mid-term (one-year) colonization process could complement or strengthen the effectiveness of any standard non-destructive methodology proposal for contributing to the assessment of GES. The objective of the present study is to evaluate the potential of ASMS for the assessment of the natural variability of assemblages on a kilometric spatial extent. This study also aims to complement the aforementioned methodology designed by Carreira-Flores et al. [38,41], by describing the succession of the ASMS-associated assemblages at two locations and over the course of one year. Two hypotheses were tested: (1) the captured variability of natural assemblages determined by ASMS will differ between locations; our experimental design is appropriate for capturing the variability in assemblages at centimetric (10s to 100s cm) and metric (10s to 100s m) scales, and we expect that the pattern will be the same on the kilometric scale (100s of km); and (2) that the captured variability of natural assemblages determined by ASMS will differ among dates.

## 2. Materials and Methods

### 2.1. Study Area

This study was carried out between June 2018 and June 2019 at two rocky reefs: Enseada de San Cristovo (ESC) (43°27′53.8″ N, 008°18′00.7″ W; 11 m in high tide) in the Ría de Ferrol and Bajo Tofiño (BT) (42°13 42.3 N, 008° 46 43.2 W; 11 m in high tide) in the Ría de Vigo (NW Iberian Peninsula) (Figure 1). The two sampling points are characterized by the presence of kelp forests of *Laminaria ochroleuca* Bachelot de la Pylaie, 1824 and *Laminaria hyperborea* (Gunnerus) Foslie, 1884, intercalated with sandbars within the rocky reef.

The Ría de Vigo is a very productive 26-km-long inlet due to the seasonal upwelling dynamics [45]. With its funnel-like morphology and SW-NE orientation, it is sheltered by the Illas Cíes against ocean waves [46]. The Ría de Ferrol exhibits a unique topography that regulates a complex current regime, resulting in a wide variety of sedimentary substrates [47]. It is also SW-NE oriented, and water exchange with the ocean is primarily controlled by tides through the narrow main channel in its middle part [48]. The outer part of the Ría de Ferrol is sheltered against ocean waves and storms by the "Porto Exterior de Ferrol". Both areas have significant human populations, with Ferrol, Narón, and Mugardos surrounding the Ría de Ferrol and totaling approximately 135,000 residents. Similarly, the Ría de Vigo is affected by high anthropic pressure, primarily from the nearly 340,000 inhabitants of Vigo, Cangas, and Moaña. These rías are impacted by human activities in the form of dockyards, commercial harbours, bivalve mollusc harvesting, sewage runoff pollution, and industrial discharge [48–50]. The Ría de Vigo also has an important area occupied by mussel raft cultures [50].

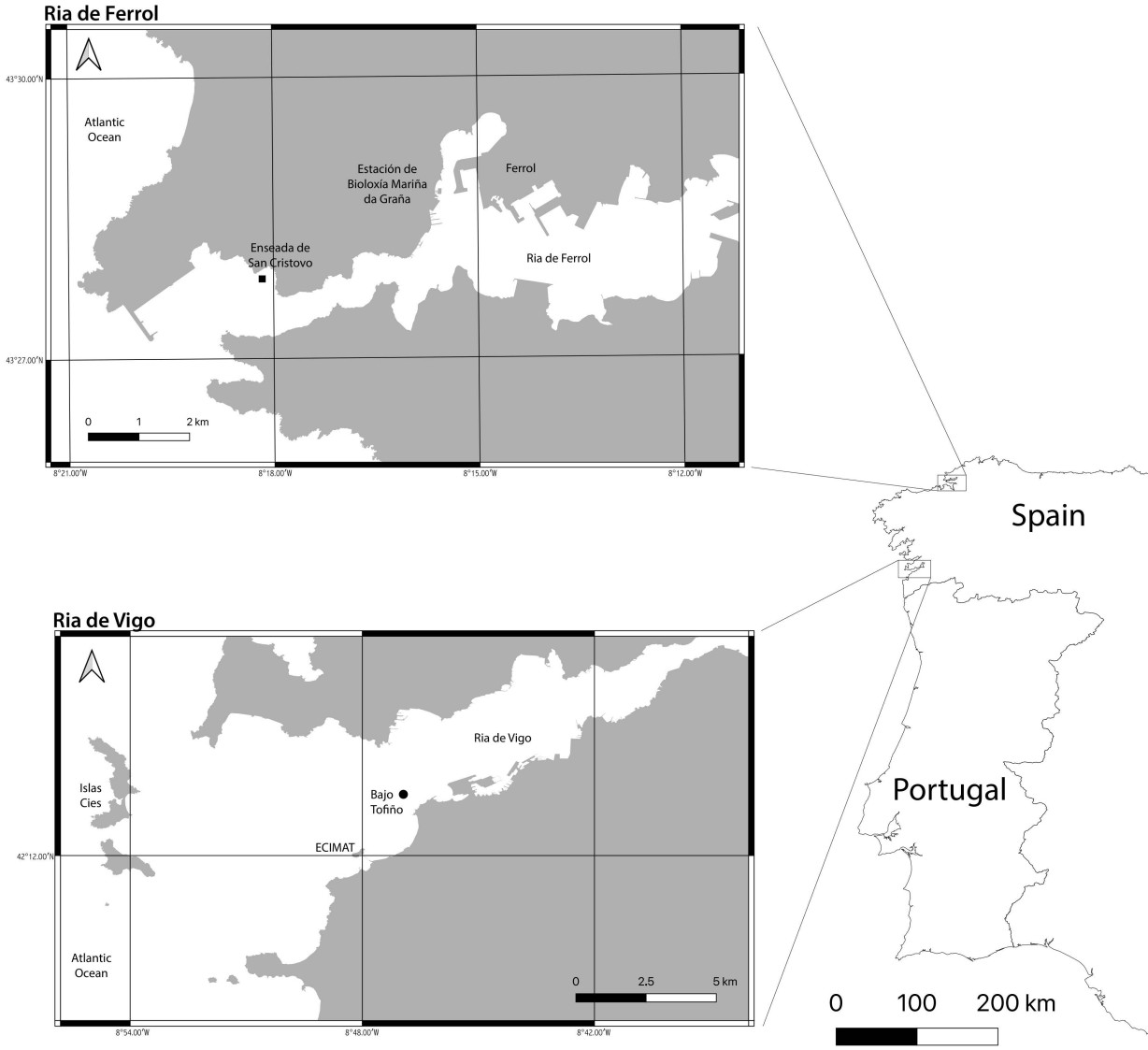

**Figure 1.** Location of study areas. ■ Enseada de San Cristovo, Ría de Ferrol; ● Bajo Tofiño, Ría de Vigo.

During the course of the study, temperature trends exhibited a similar pattern, with the lowest recorded temperatures occurring in February 2019 (12.79 ± 0.48 °C at Ría de Vigo; 12.87 ± 0.18 °C at Ría de Ferrol), while the highest temperatures were observed in July 2018 (19.46 ± 1.21 °C at Vigo; 15.54 ± 1.15 °C at Ferrol) (Figure 2). The variations in temperature during spring and summer can be attributed to differences in latitude and/or variations in the intensity of upwelling between the two locations.

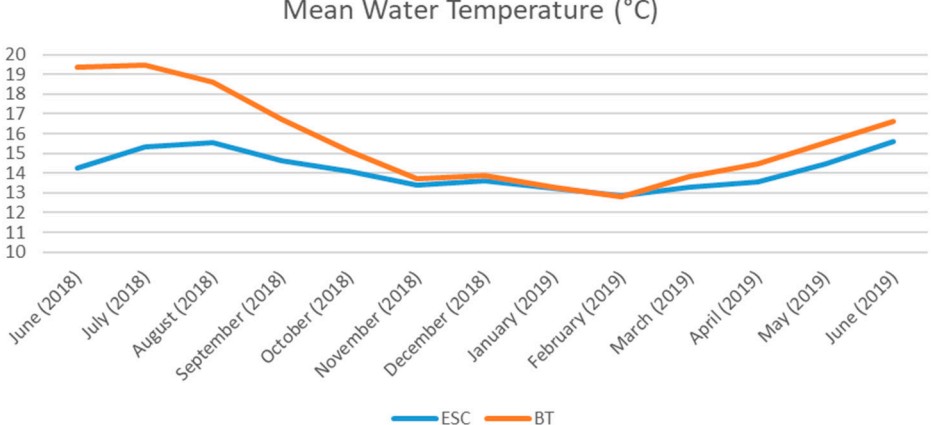

**Figure 2.** Mean temperature at Enseada de San Cristovo (ESC) and Bajo Tofiño (BT), obtained using Hobo pendant data loggers, installed at 11m in both locations.

### 2.2. Sampling Strategy

A total of forty ASMSs (Artificial Seaweed Monitoring Systems, ASMS_1 in Carreira-Flores et al. [38]) made of green polyethylene were deployed attached to 60 cm × 60 cm concrete plates within the natural settlements of macroalgae at the rocky reef (Figure 3). Specifically, twenty ASMS units were placed at ESC on 27 June 2018, and an additional twenty were placed at BT on 28 June 2018. To collect temperature data, water temperature loggers (TBI-32, Onset HOBO, Bourne, MA, USA) were attached to the concrete plates. These loggers recorded the water temperature at 5 min intervals from June 2018 to June 2019.

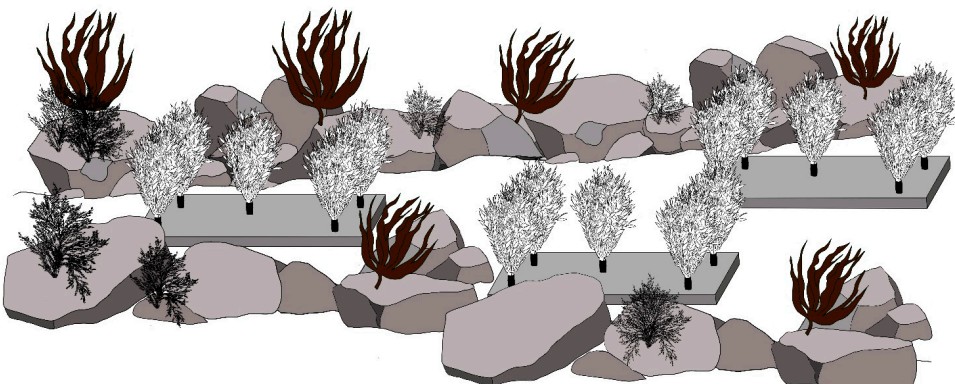

**Figure 3.** Experimental deployment of ASMS within the natural algae settlement of a rocky reef. Artificial substrates were attached to concrete plates (60 cm × 60 cm).

At both locations, a randomly selected set of five ASMS units were recovered by scuba diving after 3, 6, 9, and 12 months of deployment. The standard deployment period for ASMS units is three months, as described in previous studies [21,38,41,44]. Each substrate was carefully introduced into a 0.5 mm mesh bag and closed before being released from the base with a scraper to prevent small mobile organisms associated with the AS from escaping. Subsequently, the mesh bag was placed into a hermetic plastic bag. The associated macrofauna were washed off using filtered saltwater by shaking each AS vigorously through 0.5 mm sieves in the laboratory. The macrofauna were fixed in 99% ethanol before being quantified. Identification was performed to the species level in most cases, except when the condition of the specimen did not allow species-level identification. Taxonomic classification was performed following the World Register of Marine Species (WoRMS) [51].

*2.3. Data Analysis*

Data were analyzed using multivariate techniques to test the proposed hypotheses. The number of taxa (species richness), the total number of individuals (abundance), and the diversity (Simpson index) of the epifaunal assemblages were calculated and plotted in the R environment v 3.6.0 (Packages Vegan and Lattice) [52]. Non-parametric permutational multivariate analysis of variance (PERMANOVA; [53]) was used to test the hypotheses about differences in epifaunal assemblages. Analyses were performed based on Bray–Curtis dissimilarity matrixes from square-root-transformed density data to reduce the influence of the most abundant taxa [54]. Two hypotheses were tested: (1) the variability of natural assemblages captured using AS will differ between locations; and (2) the variability of natural assemblages captured using AS will differ among dates. For both hypotheses, the factors studied were Location (fixed, 2 levels, ESC vs. BT) and Time (random, four levels, time 1 vs. time 2 vs. time 3 vs. time 4). Although the Time factor is considered a random factor (i.e., a variance component in the model), comparisons between random factors can be allowed when there is a historical or biogeographical justification (sensu Anderson et al. [3]). These specific moments were taken into account to understand the role of time in the colonization process, assuming a 3-month period of colonization (starting at the beginning of each season), as is standard in AS colonization studies [21,38,41,44], as well as in succession studies [1,2]. When appropriate, multiple a posteriori comparisons were performed to test for differences between/within groups for pairs of levels of factors. The tests were based on 999 unrestricted random permutations of data. Additionally, non-metric multidimensional scaling (NMDS) (100 restarts) was used as the ordination method to explore differences in the assemblages' responses. Analyses of multivariate dispersion were also performed to test for the homogeneity of the dispersions between locations and dates (PERMDISP; [53]). The SIMPER procedure was used to identify each taxon's percentage contribution to the Bray–Curtis dissimilarity between the averages of groups. Taxa were considered important if their contribution to percentage dissimilarity was ≥5% and/or they contributed to explaining the first 40% (±2%) of the cumulative differences. Multivariate analyses were conducted using Primer v.6 [54] with PERMANOVA + add-on [55].

**3. Results**

Throughout this study, a total of 162 taxa and 122,822 individuals were collected (Supplementary Tables S1 and S2). At ESC, the abundance of individuals increased until the sixth month (1993.8 ± 472.5 ind/subst), declined after the ninth month (690 ± 184.8 ind/subst), and reached its maximum after the twelfth month (2724.8 ± 822.2 ind/subst) (Figure 4A). Conversely, at BT, the abundance increased at all times, reaching its peak after 12 months (8969.4 ± 470.83 ind/subst) (Figure 4A).

The Simpson index (species) was consistently higher at ESC compared to BT after 3, 6, and 9 months, with the highest value being observed after six months (0.91 ± 0.006). After 12 months, BT exhibited the greatest diversity value for temporal succession (0.89 ± 0.01), while ESC had the lowest value for this variable (0.8 ± 0.04) (Figure 4B).

Regarding species richness, at ESC, the pattern followed was similar to that of abundance, increasing and reaching its peak after six months (70.56 ± 11.36), decreasing after nine months (54 ± 8.7), and experiencing growth after twelve months (66.2 ± 7.85). At BT, species richness increased after six months (60.8 ± 10.32), remained stable after nine months (57 ± 12.3), and reached its maximum level after twelve months (73.4 ± 6.42) (Figure 4C).

The PERMANOVA results for the composition of the harbored assemblages showed a significant interaction between Location x Time (Table 1). PERMDISP analyses showed that these differences were due to the distance of the centroids rather than data dispersion (Table 1). Pairwise tests revealed significant differences in macrofaunal assemblages between locations at each time point (Table 2). These patterns were also evident in the NMDS plot (Figure 5), showing that the AS captured differences in environmental variability at the two locations over time.

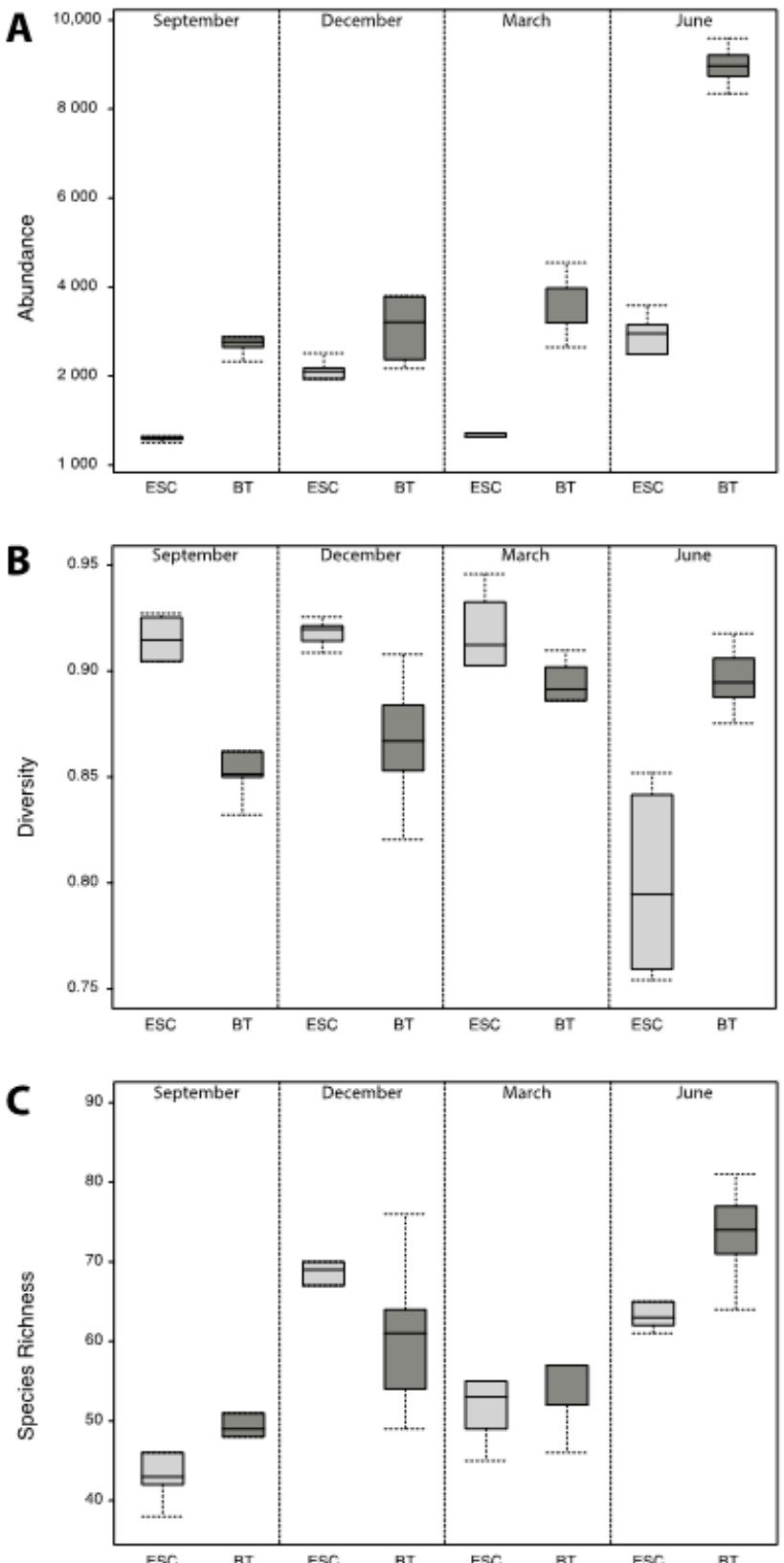

**Figure 4.** (**A**) Total abundance per substrate (number of individuals); (**B**) species (Simpson index), and (**C**) species richness per substrate (number of taxa) of macrofaunal assemblages associated with ASMS at Enseada de San Cristovo (ESC) and Bajo Tofiño (BT) at each data collection (3 months: September 2018; 6 months: December 2018; 9 months: March 2019; 12 months: June 2019).

**Table 1.** Summary of PERMANOVA results for total assemblages at ESC and BT.

| Source | df | MS | Pseudo-F | P (perm) | Unique Perms |
|---|---|---|---|---|---|
| Location | 1 | 21603 | 6.4063 | 0.002 | 579 |
| Time | 3 | 5373.7 | 21.164 | 0.001 | 997 |
| Location × Time | 3 | 3372.1 | 13.281 | 0.001 | 999 |
| Residual | 32 | 253.91 | | | |
| Total | 39 | | | | |
| Permdisp | | | P (perm):0.371 | | |

**Table 2.** Results of pair-wise test between ASMS at Enseada de San Cristovo (ESC) and Bajo Tofiño (BT) at each data collection (3 months: September 2018; 6 months: December 2018; 9 months: March 2019; 12 months: June 2019). *, $p < 0.05$; **, $p < 0.01$. Number of unique perms = 126 for all taxonomic levels and combinations.

| | 3 Months | 6 Months | 9 Months | 12 Months |
|---|---|---|---|---|
| | t | t | t | t |
| ESC vs. BT | 5.0261 * | 5.5261 * | 5.3744 * | 6.6794 ** |

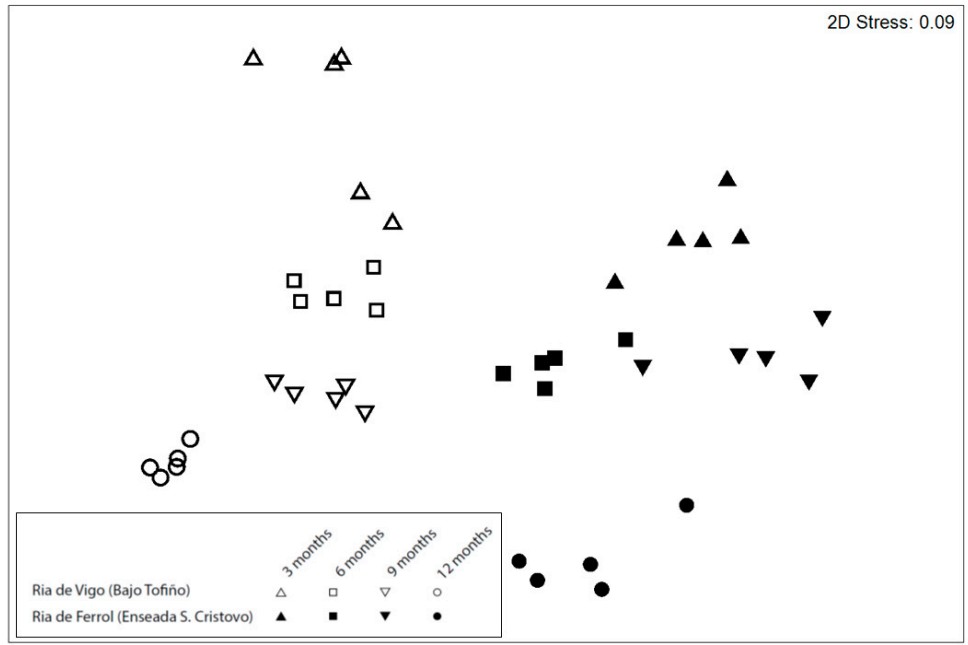

**Figure 5.** Non-metric multidimensional scaling (nMDS) ordination based on Bray–Curtis dissimilarity measures of macrofaunal assemblages density data of ASMS at Enseada de San Cristovo (ESC) and Bajo Tofiño (BT) at each data collection (3 months: September 2018; 6 months: December 2018; 9 months: March 2019; 12 months: June 2019).

After 3 months, both locations shared 43 taxa, but ESC supported 27 exclusive taxa, while BT had 26 exclusive taxa (Table 3). The dissimilarity between ES and BT was 59.37%, with six amphipod species being the taxa responsible for 40.49% of this dissimilarity (Table S3).

**Table 3.** Number of common and exclusive taxa of AS at Enseada de San Cristovo (ESC) and Bajo Tofiño (BT) at each data collection (3 months: September 2018; 6 months: December 2018; 9 months: March 2019; 12 months: June 2019).

|  | 3 Months | 6 Months | 9 Months | 12 Months | Total |
|---|---|---|---|---|---|
| Common | 43 | 56 | 52 | 67 | 100 |
| Only ESC | 27 | 43 | 24 | 30 | 43 |
| Only BT | 26 | 27 | 35 | 26 | 19 |
| Total ESC | 70 | 99 | 76 | 95 | 143 |
| Total BT | 69 | 83 | 87 | 91 | 119 |

After 6 months, the two locations shared 56 taxa, but ESC supported 43 exclusive taxa versus the 27 exclusive taxa at BT (Table 3). The dissimilarity between ESC and BT was 55.47%, with eleven species of amphipod being the taxa responsible for 41.4% of the dissimilarity between locations (Table S3).

After 9 months, the two locations shared 52 taxa, but ESC supported 24 exclusive taxa versus the 35 exclusive taxa at BT (Table 3). The dissimilarity between ESC and BT was 61.82%, with eight amphipod species being the taxa responsible for the 40.28% dissimilarity between locations (Table S3).

After 12 months, the two locations shared 67 taxa, but ESC supported 30 exclusive taxa versus 26 exclusive taxa at BT (Table 3). The dissimilarity between locations was high (61.94%). The dissimilarity between locations was high (61.94%), with five amphipod species, one equinoderms species, and two annelids being the taxa responsible for 41.90% of the dissimilarity between the locations (Table S3).

Regarding changes in assemblages over time, the PERMANOVA results showed significant differences regarding the Time factor (Table 1). The previous pattern could be observed at the MDS (Figure 5).

At ESC, after 3 months, amphipods accounted for 52.44% of the total individuals, followed by decapods (11.82%) and sabellids (10.17%) (Figures S1–S3). After 6 months, the percentages of amphipods and sabellids remained stable, accounting for 51.74% and 9.74%, respectively (Figures S1–S3). After 9 months, the percentage of amphipods started to decrease, accounting for 42.50%, followed by sabellids (11.86%) (Figures S1–S3). Finally, after 12 months, sabellids accounted for 51.69% of the total individuals, followed by amphipods (19.78%) and decapods (15.13%) (Figures S1–S3). Between 3 and 6 months, seven species of amphipod and two species of polychaetes contributed to 40.50% of the dissimilarity at ESC (average dissimilarity = 47.27%) (Table S4). The dissimilarity between the 6-month and 9-month samples was 41.46%, with seven species of amphipod, two species of bivalve, one species of gastropod, and one species of annelid contributing to 41.68% of the dissimilarity (Table S4). The dissimilarity between the 9-month and 12-month samples was 48.93%, with three species of annelid, four species of amphipod, and one species of decapod comprising the taxa responsible for 41.16% of this dissimilarity (Table S4).

At BT, after 3 and 6 months, amphipods were the most abundant group, accounting for 84.92% and 80.91% of the total individuals, respectively (Figures S1–S3). After 9 months, the relative abundance of amphipods decreased to 74.93%, followed by sabellids (5.06%) and decapods (4.70%) (Figures S1–S3). After 12 months, the relative abundance of amphipods reached its minimum value, accounting for 49.24 %, followed by holoturoids (23.35%) and decapods (8.88%) (Figures S1–S3). At BT, six species of amphipod contributed to 39.47 % of the dissimilarity between the 3-month and 6-month samples (average dissimilarity: 39.50%) (Table S5). The taxa responsible for 39.34% of the dissimilarity between the 6-month and 9-month samples (average dissimilarity: 36.15) consisted of eight species of amphipod (Table S5). Between the 9-month and 12-month samples (average dissimilarity: 38.40), the taxa responsible for 40.91% of the dissimilarity consisted of two species of echinoderm, one species of bivalve, one species of decapod, one genus of annelid, and four species of amphipods (Table S5).

## 4. Discussion

Our results showed that the initial hypotheses were supported. The ASMS effectively captured the natural variability of the assemblages on a kilometric scale and over different deployment times, revealing differences in the epifaunal assemblages between locations and under similar conditions, as well as across different dates.

It can be assumed that a mostly local process is responsible for the availability of colonizing animals for ASMS (sensu Ref. [56]). Previous studies have demonstrated that in the early stages of deployment, the AS is colonized by most of the elements making up the mobile invertebrate fauna in their nearby area [31,56,57], which is sensitive to local variations and environmental conditions [32,57]. Our findings are in agreement with the work of Carreira-Flores et al. and Norderhaug et al. [34,38,41], who reported that in the early stages of colonization, complex artificial three-dimensional structures are predominantly colonized by peracarid amphipods, reflecting the horizontal dispersal patterns and mobility capabilities of these organisms. Conversely, in non-complex AS, such as the PVC plates commonly used in colonization studies in sessile epifauna [19], amphipods are not the most efficient colonizers during the early stages [1]. In non-complex structures, the development of complexity relies on sessile habitat-forming organisms to create suitable conditions for hosting high abundances of amphipods, which significantly prolongs the colonization process. This suggests that, despite the high mobility of amphipods, they exhibit a stronger affinity for higher structural complexity [58], making more complex AS more effective at capturing these early colonizers. Our results indicate that peracarid amphipods are the primary representatives responsible for the variability observed in the ASMS between different locations and dates. We observed an increasing trend in the number of colonizing amphipods, highlighting their dominant role in marine epifaunal assemblages and their successful colonization of complex AS [59,60]. However, while the number of amphipods increased over time, their relative abundance (considering the total number of individuals captured across all species) declined after 6 months of deployment, reaching its lowest value after 12 months at both locations. Conversely, the percentage of other taxa with pelagic larvae recruitment cycles, such as sessile polychaetes like *Filograna implexa* Berkeley, 1835, *Spirobranchus triqueter* (Linnaeus, 1758), and *Janua heterostropha* (Montagu, 1803), increased over time. This shift in the relative abundance from species with adult motile dispersal abilities ("early colonizing species") to species with pelagic larvae recruitment strategies ("late colonizing species") suggests an ongoing maturation process in the community [61]. Our findings also align with the observations of Underwood and Chapman [18], who reported variations and replacements of taxa over time. The use of complex ASMS in our methodology enabled the acquisition of accurate data regarding the status and composition of surrounding assemblages in the short term. ASMS are effectively able to capture a wide variety of amphipods, which are a dominant component of marine epifaunal assemblages, and also exhibit sensitivity to the arrival of recruited species in the medium term.

Comparing the abundance, species richness, and Simpson indexes with the results published by Carreira-Flores et al. [38,41], we did not observe a comparable trend on any of the examined dates. Specifically, for the ASMS deployed at ESC, there were no similarities in any of the aforementioned indexes when comparing the results after 3 months of deployment (March 2018 vs. September 2018) or 6 months (June 2018 vs. December 2018), or when comparing the same months in different years (March 2018 vs. March 2019; June 2018 vs. June 2019). This lack of similarity supports the findings of García-Sanz et al. [33], who suggested that even the same type of substrate can yield different results depending on the deployment time and the natural yearly variability. These differences can be attributed to variations in recruitment periods, the life cycle of the surrounding benthic fauna, or the presence of "early" and "late" colonizing fauna [1]. For instance, the NW Iberia region is characterized by pronounced seasonality and upwelling events that synchronize larval recruitment during spring and early summer [62,63]. Consequently, differences in deployment period could explain the variations observed when using the same standard ASMS structures. Our proposed non-destructive sampling methodology

is sensitive to the natural variability of assemblages, regardless of their deployment date and period. This highlights the importance of standardizing the deployment and recovery periods for ASMS to avoid introducing biases in studies and to ensure the comparability of data. By establishing consistent timeframes, the potential biases arising from differences in deployment and the duration before recovery can be minimized.

The ASMS assemblage analysis revealed significant differences in the epifaunal assemblages across all sampling dates and locations, probably indicating the influence of inherent variability within the natural assemblages at each location. Our findings support the hypothesis that benthic assemblages do not exhibit linear dynamics, and must be understood to be a mixture of different successional stages [2,25] that change over time [64], driven by each site's unique characteristics and natural variability. At ESC, the abundance (total number of individuals) and species richness did not increase linearly. Instead, it exhibited two peaks: one in December after 6 months of deployment and another in June after 12 months. Similarly, at BT, species richness displayed a similar trend, while the abundance of individuals remained relatively stable after 3, 6, and 9 months, with a notable peak observed after 12 months of deployment. Satterthwaite et al. [65] reported variations in larval assemblage abundance and composition influenced by upwelling and relaxation dynamics, affecting recruitment cycles. Upwelling events, altering primary productivity by increasing nutrient availability, and influencing larval dynamics in the Galician Rías, could impact the colonization and succession processes of ASMS. Underwood and Chapman [18] suggested that succession did not stabilize within 6 months of deployment in their study. Consistent with their findings, our study did not identify a period in which the community stabilized, as evidenced by the significant differences observed between each dataset at both locations. This indicates that the climax stage may not have been reached after one year of deployment. Furthermore, Zupan et al. [66] found that a climax state was not achieved even after 11 years in large AS, highlighting the dynamic nature of marine ecosystems and the potential absence of an orthodox definition of climax. Consequently, to gain a comprehensive understanding of colonization dynamics in complex AS over the long term, extended deployment periods should be considered.

In light of there being some environmental factors that were common to both of the selected deployment locations (e.g., same depth, same surrounding macroalgae communities, comparable protection against waves, and comparable anthropic pressure), the differences in the assemblages observed between the deployment locations can be attributed to other drivers, with hydrodynamics and the latitudinal gradient being plausible explanatory factors. These drivers, individually or in combination, also influence the variability of the sampled assemblages captured by the ASMS, as evidenced by the distinct assemblages and numerous exclusive species at each site. Subtle changes in hydrodynamic conditions can regulate the epifaunal community, according to Conradi et al. [67], and the exclusive appearance of *Stenothoe monoculoides* (Montagu, 1813) and the greater abundance of *Ericthonius brasiliensis* (Dana, 1853) at BT suggests that, despite both locations "a priori" having the same features, BT may have been subjected to higher hydrodynamic conditions. Another plausible explanation for the differences in the assemblages is related to the latitudinal difference between the sites, with ESC being located at 41° and BT at 42° latitude N. These locations are subjected to different intensities of upwelling events, which, as mentioned earlier, can explain the differences in summer temperatures. Many species exhibit ecological variations across latitudes in response to large-scale environmental variability [68]. In the case of the Iberian Peninsula, latitudinal differences have been documented for oceanographic patterns (e.g., chlorophyll, water temperature, nutrient availability due to spatial variations in upwelling intensity and frequency), as well as macroalgae and fish assemblages [69]. Therefore, expecting a latitudinal gradient in the epifaunal assemblages captured by the ASMS is reasonable. Alternatively, the observed assemblage differences could be attributable merely to local inherent variability. Regardless, the assemblages captured by the ASMS were sensitive to the consequences of these different drivers, indicating that monitoring methodologies employing ASMS are responsive to large-scale variables.

Further studies encompassing a broader geographic and temporal range are necessary to elucidate the drivers underlying the observed assemblage differences and determine whether the latitudinal gradient, hydrodynamic conditions, or local intrinsic variability do indeed influence these differences.

## 5. Conclusions

Developing standardized and replicable monitoring protocols is essential for accurately describing species composition and assessing the seabed's GES. To achieve this, it is crucial to validate the effectiveness of candidate methodologies at capturing the natural variability of assemblages at different spatial and temporal scales. This study complements and reinforces the non-destructive standard methodology proposed by Carreira-Flores et al. for use in benthic monitoring studies, and can be applied irrespective of the geographic location. As highlighted in our previous works, the proposed methodology is suitable for capturing variability in assemblages at the centimeter (10s to 100s of centimeters) and meter (10s to 100s of meters) scales [38,41]. The results of this study further demonstrate its effectiveness at the kilometric scale (100s of km). Moreover, the results are in agreement with the observations made by Carreira-Flores et al. [38], confirming that this methodology can effectively capture the natural variability of assemblages and is able to distinguish different successional stages, enabling the tracking of the entire succession process. Finally, another key point is the requirement of standardizing deployment and recovery periods for the ASMS, ensuring total comparability of the data.

**Supplementary Materials:** The following supporting information can be downloaded at: https://www.mdpi.com/article/10.3390/d15060733/s1, Table S1. Number of individuals (average ± standard deviation) captured at Enseada de San Cristovo (ESC) and Bajo Tofiño (BT) in September 2018, 3 months (S) and December 2018, 6 months (D). Table S2. Number of individuals (average ± standard deviation) captured at Enseada de San Cristovo (ESC) and Bajo Tofiño (BT) in March 2019, 9 months (M) and June 2019, 12 months (J). Table S3. Contribution (%) of individual taxa and cumulative percentage (Cum %) from ASMS assemblages to the average Bray–Curtis dissimilarity between Enseada de San Cristovo (ESC) and Bajo Tofiño (BT) at each data collection (3 months: September 2018; 6 months: December 2018; 9 months: March 2019; 12 months: June 2019). Table S4. Contribution (%) of individual taxa and cumulative percentage (Cum %) from ASMS assemblages to the average Bray–Curtis dissimilarity at each data collection (3 months: September 2018; 6 months: December 2018; 9 months: March 2019; 12 months: June 2019) at Enseada de San Cristovo (ESC). Table S5. Contribution (%) of individual taxa and cumulative percentage (Cum %) from ASMS assemblages to the average Bray–Curtis dissimilarity at each data collection (3 months: September 2018; 6 months: December 2018; 9 months: March 2019; 12 months: June 2019) at Bajo Tofiño (BT). Figure S1. Total assemblage composition (%) of ASMS at Order-Level of Enseada de San Cristovo (ESC) and Bajo Tofiño (BT) at every time point (S: September 2018, 3 months; D: December 2018, 6 months; M: March 2019, 9 months; J: June 2019, 12 months). Figure S2. Amphipod assemblage composition (%) of ASMS s at family level of Enseada de San Cristovo (ESC) and Bajo Tofiño (BT) at every time point (S: September 2018, 3 months; D: December 2018, 6 months; M: March 2019, 9 months; J: June 2019, 12 months). Figure S3. Amphipod assemblage composition (%) of ASMS s at species level of Enseada de San Cristovo (ESC) and Bajo Tofiño (BT) at every time point (S: September 2018, 3 months; D: December 2018, 6 months; M: March 2019, 9 months; J: June 2019, 12 months).

**Author Contributions:** Conceptualization, E.C., G.D.-A. and P.T.G.; methodology, D.C.-F., R.N., H.R.S.F. and G.D.-A.; formal analysis, D.C.-F., G.D.-A. and M.R.; investigation, D.C.-F., R.N., H.R.S.F., G.D.-A., P.T.G.; resources, G.D.-A. and P.T.G.; data curation, D.C.-F., M.R. and E.C.; writing—original draft preparation, D.C.-F.; writing—review and editing, D.C.-F., M.R. and P.T.G.; supervision, E.C., G.D.-A. and P.T.G.; funding acquisition, P.T.G. All authors have read and agreed to the published version of the manuscript.

**Funding:** This study was supported by the project ATLANTIDA (ref. NORTE-01-0145-FEDER-000040), supported by the Norte Portugal Regional Operational Programme (NORTE 2020), under the PORTUGAL 2020 Partnership Agreement and through the European Regional Development Fund (ERDF). This work was also supported by the "Contrato-Programa" UIDB/04050/2020 funded by national funds through the FCT I.P.

**Institutional Review Board Statement:** Not applicable.

**Data Availability Statement:** The data presented in this study are available on request from the corresponding author. The data are not publicly available because this data set may be included as part of other ongoing studies.

**Acknowledgments:** We would like to thank all the members of the Marine Biology Station of A Graña (EBMG) and the Toralla Marine Science Station (ECIMAT) for providing the Stations' resources and for their assistance in the verification of the identification process. We would also like to acknowledge the two anonymous reviewers, as well as the editor, for their valuable comments.

**Conflicts of Interest:** The authors declare that they have no conflict of interest. The funders had no role in the design of the study; in the collection, analyses, or interpretation of data; in the writing of the manuscript, or in the decision to publish the results.

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
