# Peer review of "Colonization in Artificial Seaweed Substrates: Two Locations, One Year"

_diversity, doi:10.3390/d15060733_

Round 1
Reviewer 1 Report
The present work investigated the epifauna succession in artificial substrates by comparing two latitudes during one year. The results are interesting for attracting readers. However, there are some suggestions for the authors’ reference:
1. I wonder if this methodology is appropriate to capture assemblages to imitate the natural sediment, at the conclusion, the author stated this. I suggest the authors disscuss it in detail because this is very important.
2. The two sampling sites are different in hydrodynamic condition and also other environmenal difference. I do not think the comparation between them are meaningful.
3. To investigate the succession need a long time period. It is better to compare the colonization process by same seasnon, exceptive of the sampling site without seasonal changes in temperature because the species composition are very different with seasons.
4. I strong suggest the authors provides a figure to show the changes of species composition with times both in different season in one year and same season in different year.
Author Response
- I wonder if this methodology is appropriate to capture assemblages to imitate the natural sediment, at the conclusion, the author stated this. I suggest the authors disscuss it in detail because this is very important.
We understand the referee´s concern. However, the objective of this paper was not to mimic natural sediment to capture macrobenthic assemblages and this is not discussed anywhere in the paper. For sampling sediments, better approaches can be used than the ASMS described, such as van been grab or corers. Instead, building upon previous studies, the approach used here involves utilizing three-dimensional complex structures that can capture the natural variability over time at different spatial scales. As stated on page 2, lines 86-89, “It has also been proved that AS are a valid tool to distinguish macroalgae-like and crevice-like macrofauna assemblages from distinctive locations and throughout different time scales [19,39,42]" and further reinforced in the Conclusions section, where it is mentioned: “This study complements and reinforces the strengths of the non-destructive standard methodology proposed by Carreira-Flores et al. (2020,2021) for benthic monitoring studies that can be used regardless of the geographic location”. In our previous papers, we extensively discuss the ability of ASMS to replicate natural dendritic and crevice habitats to capture natural biodiversity. Therefore, we believe that discussing the statements of previous papers again does not contribute with anything new. Hence, reinforcing the strengths of the methodology and discussing the colonization process of the ASMS is the goal of this paper.
- The two sampling sites are different in hydrodynamic condition and also other environmenal difference. I do not think the comparation between them are meaningful.
We understand the referee´s concern about the comparison of two sampling points. In this sense, for this paper we selected two locations in two different Rias with the objective of asses the capacity of AS of capturing natural variability, expecting that ASMS will harbor different assemblages being aware of the expected differences caused by different latitudes, upwelling intensity/duration, inherent local variability or hydrodynamic conditions of the sampling sites. We consider that the comparison of both sampling sites is a fundamental axis to explain the assemblages differences to reinforce the capacity of the methodology to be sensitive to natural variability due to environmental changes.
- To investigate the succession need a long time period. It is better to compare the colonization process by same seasnon, exceptive of the sampling site without seasonal changes in temperature because the species composition are very different with seasons.
Despite it would be very interesting to have long-term colonization data, the main goal of the methodology is to be an alternative to capture natural biodiversity variability for monitoring programs minimizing the deployment periods in order to minimize the possibilities of loss or destruction of the AS caused by vandalism or adverse environmental conditions. On the other hand, for direct comparisons between seasons, another experimental design that considers the deployment of ASMS in the initial days of each season and the recovery after 3 months at the end of each season needs to be implemented. This experimental design that takes into account the deployment of “clean” ASMS at the beginning of each season will guarantee the non-interaction of previous colonizers in the colonization process, in this sense, and as it is published in several papers, some authors have demonstrated that the period, deployment time and background colonization of the AS have an extensive effect in the succession process. However, the AS compared in this study were deployed all at the same time, eliminating the confounding factor of seasonality. Once the confounding factor of the season is eliminated, the objective of this study was to explore the colonization and the ability of AS to capture natural variability on benthic assemblages. Finally, if the experiment were deployed on another season the species on assemblages would be different but differences between localities are expected to remain.
- I strong suggest the authors provides a figure to show the changes of species composition with times both in different season in one year and same season in different year.
We agree with the referee´s suggestion, the figure of the composition of the assemblage is now included in the supplementary material.
Reviewer 2 Report
Comments on the typescript ‘Succession in Artificial Substrates: two latitudes, one year’ submitted by Diego Carreira-Flores et al. to Diversity
The paper evaluates the potential of Artificial Structure deployed at 11 m depth to describe the assemblages´ natural variability on a spatial extent of more than 200 km, associated with the macrofauna at two Rías of the Galician Coast (NW Iberian Peninsula) after 3, 6, 9, and 12 months after implementation. The results showed that macrofauna differed between and within locations at every data.
The paper is well written and described the short-term colonisation during one year of such AS in temperate water. The AS appeared mainly colonised by vagile species mainly amphipods which ich were able to occupy rapidly these artificial habitats. The fauna showed different colonisation process in both rias but the number of common species was high: 100 on a total of 169 (60%) (Table 4). Probably main of the species (taxa) had sampled in few specimens.
The paper should be published after minor corrections.
Title It is not really a succession but a colonisation process: Artificial Structures in two Rias from Galicia (Spain): one year of colonisation
Page 2, line 2 to 19. This paragraph needs to be reduced. Line 27, precise the term slow and over a long-time scale, in the page 3 the term mid-term colonization should be also defined. Last paragraph, conversely the AS shows generally different fauna than natural habitat, and the natural habitat was more complex than artificial structure.
Page 3. 11 m depth: at low tide, high tide?
Page 4. Figure 1, explain the difference of temperature during the summer between ESC the colder and BT the warmer.
Page 5. Where the ASMS were placed on 27 and 28 June 2018?
The part on results (pages 8 and 9) could be shortened and referred to the table in supplementary data.
In spite that the data will be included in other studies, it was necessary to give the list of taxa found in both rias with the number of specimens collected.
Page 11. The colonisation should be different if the AS were deployed in summer, autumn and summer?
I do not agree with the term climax. What do you mean with permanent stable climax, all the ecosystem are dynamic and changes permanently.
Given the similar environmental factors… no the temperature is lower in ESC in summer.
Taxonomic remarks
Harmotoe = Harmothoe
Aora gracillis = Aora gracilis
Author Response
Title It is not really a succession but a colonisation process: Artificial Structures in two Rias from Galicia (Spain): one year of colonisation
According to the referee´s suggestion, we have rearranged the title: Colonization in Artificial Seaweed Substrates: two locations, one year
Page 2, line 2 to 19. This paragraph needs to be reduced. Line 27, precise the term slow and over a long-time scale, in the page 3 the term mid-term colonization should be also defined. Last paragraph, conversely the AS shows generally different fauna than natural habitat, and the natural habitat was more complex than artificial structure.
According to the referee´s suggestion, we noticed that the text was confusing so we have modified the text in page 2 and page 3.
Page 3. 11 m depth: at low tide, high tide?
According to the referee´s suggestion, we have included the information in page 3 line 129.
Page 4. Figure 1, explain the difference of temperature during the summer between ESC the colder and BT the warmer.
According to the referee´s suggestion, we have explained the temperature difference: “Differences on temperature during spring and summer may be due to latitudinal differences and/or different intensity of the upwelling between localities”. However, the discussion of the responsible of this difference is out of the scope of this paper.
Page 5. Where the ASMS were placed on 27 and 28 June 2018?
This information is already in the manuscript (page 5, lines 158-161): “Twenty AS were placed at ESC on June 27, 2018, and twenty AS on June 28, 2018, at BT.” June 27 in Enseada de San Cristovo (ESC, Ria de Ferrol ) and June 24 Bajo Tofiñi (BT, Ria de Vigo)
The part on results (pages 8 and 9) could be shortened and referred to the table in supplementary data.
According to the referee´s suggestion, we have shortened the results
In spite that the data will be included in other studies, it was necessary to give the list of taxa found in both rias with the number of specimens collected.
According to the referee´s suggestion, we have included a table in the supplementary materials
Page 11. The colonisation should be different if the AS were deployed in summer, autumn and summer?
The deployment season will probably affect the structure of assemblages found in the AS. In order to avoid this confounding effect all the ASMS were deployed at the same time of the year. The effect of the season on the colonization of the AS is a very interesting topic but, it was not considered in this study (See also response to referee 1 about this issue)
I do not agree with the term climax. What do you mean with permanent stable climax, all the ecosystem are dynamic and changes permanently.
According to the referee´s suggestion, we have modified the term because it is confusing.
Given the similar environmental factors… no the temperature is lower in ESC in summer.
According to the referee´s suggestion, we have clarified this section
Taxonomic remarks
Harmotoe = Harmothoe
Aora gracillis = Aora gracilis
According to the referee´s suggestion, we have corrected the errors
Round 2
Reviewer 1 Report
The authors have revised the MS according to the suggestions of reviewers. It now can be accepted after minor revision.
Author Response
We are glad that the changes have been to the referee's liking. Finally, thank you for your suggestions as they have been very beneficial.